# Synthesis of Novel *N*-Heterocyclic Carbene-Ruthenium (II) Complexes, “Precious” Tools with Antibacterial, Anticancer and Antioxidant Properties

**DOI:** 10.3390/antibiotics12040693

**Published:** 2023-04-01

**Authors:** Jessica Ceramella, Rubina Troiano, Domenico Iacopetta, Annaluisa Mariconda, Michele Pellegrino, Alessia Catalano, Carmela Saturnino, Stefano Aquaro, Maria Stefania Sinicropi, Pasquale Longo

**Affiliations:** 1Department of Pharmacy, Health and Nutritional Sciences, University of Calabria, Via P. Bucci, 87036 Arcavacata di Rende, Italy; jessica.ceramella@unical.it (J.C.); domenico.iacopetta@unical.it (D.I.); michele.pellegrino@unical.it (M.P.); stefano.aquaro@unical.it (S.A.); 2Department of Chemistry and Biology, University of Salerno, Via Giovanni Paolo II, 132, 84084 Fisciano, Italy; rutroiano@unisa.it (R.T.); plongo@unisa.it (P.L.); 3Department of Science, University of Basilicata, Viale dell’Ateneo Lucano 10, 85100 Potenza, Italy; carmela.saturnino@unibas.it; 4Department of Pharmacy-Drug Sciences, University of Bari “Aldo Moro”, 70126 Bari, Italy; alessia.catalano@uniba.it

**Keywords:** *N*-heterocyclic carbenes, human topoisomerases, breast cancer cells, ruthenium complexes

## Abstract

Ruthenium *N*-heterocyclic carbene (Ru-NHC) complexes show interesting physico-chemical properties as catalysts and potential in medicinal chemistry, exhibiting multiple biological activities, among them anticancer, antimicrobial, antioxidant, and anti-inflammatory. Herein, we designed and synthesized a new series of Ru-NHC complexes and evaluated their biological activities as anticancer, antibacterial, and antioxidant agents. Among the newly synthesized complexes, **RANHC-V** and **RANHC-VI** are the most active against triple-negative human breast cancer cell lines MDA-MB-231. These compounds were selective in vitro inhibitors of the human topoisomerase I activity and triggered cell death by apoptosis. Furthermore, the Ru-NHC complexes’ antimicrobial activity was studied against Gram-positive and -negative bacteria, revealing that all the complexes possessed the best antibacterial activity against the Gram-positive *Staphylococcus aureus*, at a concentration of 25 µg/mL. Finally, the antioxidant effect was assessed by DPPH and ABTS radicals scavenging assays, resulting in a higher ability for inhibiting the ABTS^•+^, with respect to the well-known antioxidant Trolox. Thus, this work provides encouraging insights for further development of novel Ru-NHC complexes as potent chemotherapeutic agents endowed with multiple biological properties.

## 1. Introduction

Among precious metals, ruthenium (Ru) has unique physico-chemical properties, widely evaluated by numerous research groups worldwide for about 40 years, which makes it particularly useful in drug design [1]. Ru is a good catalyzer, as demonstrated by the Nobel prizes in 2001 (Noyori, enantioselective hydrogenation reactions catalyzed by chiral Ru complexes) [2] and 2005 (Grubbs, olefin metathesis reactions catalyzed by Ru carbene complexes) [3]. Ru holds physico-chemical properties similar to those of the iron family in the periodic table, which allow binding diverse biological macromolecules and a low impact on normal cells [4,5]. Notably, Ru complexes also produce negligible systemic toxicity [6,7], good antitumor activity, anti-angiogenic and antimicrobial properties [8,9,10,11]; and, more recently, antioxidant and anti-inflammatory activities have been reported [12,13]. Generally, *N*-heterocyclic carbene (NHC) complexes with noble metals, including ruthenium, gold and silver, have shown interesting biological activities, particularly as anticancer and antimicrobials [14,15,16]. In this context, our research group contributed with in vitro studies on gold and silver NHC complexes particularly active against breast cancer and cervix cell lines, and several interesting results were obtained [17]. Additionally, we have recently reviewed the importance of Ru-*N*-heterocyclic-carbene (Ru-NHC) complexes for their biological actions [18]. Some Ru-based coordination compounds have already entered clinical trials [19,20,21,22] and, recently, a new class of complexes, different from those already under clinical development, has been introduced. These complexes are based on Ru(II), which is stabilized by a coordinated η^6^-arene. Among the latter, the RAPTA family (R = ruthenium, A = arene and PTA = 1,3,5-triaza-7-phosphaadamantane) ([Ru( η^6^-arene)(PTA)X_2_] is characterized by a piano stool structure in which Ru has a coordinated η^6^-arene ligand, the amphiphilic phosphine ligand PTA and two chlorides, which are the labile ligands. The PTA ligand is the one that characterizes the RAPTA structure and can improve the water solubility of these complexes. The first structure of this class of complexes reported is that of [Ru(η^6^-p-cymene)Cl_2_(PTA)] [23,24], called RAPTA-C (Figure 1).

In the present paper, we aimed to investigate the anticancer, antimicrobial, and antioxidant properties of a new synthesized series of mononuclear Ru-NHC complexes stabilized by η^6^-arene coordinated ligand (named RANHC, see Figure 1). NHCs can effectively replace phosphines, owing to the easy regulation of the steric and electronic properties of these auxiliary ligands. NHCs generally have more σ-electrodonor properties than phosphines, and their complexes are thermodynamically very stable. This is due to the strong metal-carbon bonding, by σ-donation of the sp^2^ lone pair of the carbene carbon atoms to one empty orbital transition metal [25].

The anticancer activity of the new complexes was studied in two breast cancer cell lines, MCF-7 and MDA-MB-231, and the neuroblastoma cells SH-SY5Y. Complexes **RANHC-V** and **-VI** were the most active overall and, in particular, towards the MDA-MB-231 breast cancer cells. These complexes were found to be selective inhibitors of human topoisomerase I, an essential DNA-dependent enzyme catalyzing changes in the topological state of DNA and involved in replication, transcription, and other vital cell metabolisms [26]. Additionally, the newly synthesized Ru-NHC complexes were endowed with antibacterial activity, in particular, against *S. aureus* Gram-positive, with a MIC of 25 µg/mL, and among all, **RANHC-III** and **-VI** exhibited a wider spectrum. Moreover, they exhibited significant ABTS scavenging activity, mostly the complex **RANHC-III**, whose IC_50_ value is about 17-fold lower than that obtained for Trolox. Summing up, our findings indicate that, among the novel Ru-NHC complexes, **RANHC-III** exhibited the best antioxidant and antimicrobial activity, **RANHC-V** and **-VI** possessed the best anticancer activity against MCF-7 and MDA-MB-231 breast cancer cells, together with good antioxidant and antimicrobial activity. Overall, these outcomes suggest tremendous therapeutic potential of the studied molecules to be exploited for development of new effective anticancer, antibacterial, and antioxidant agents.

## 2. Results and Discussion

### 2.1. Chemistry

The new Ru complexes **RANHC-I, -II, -III, -IV, -V, -VI** (Figure 2) were synthesized according to the procedure reported in previous studies [27,28,29] as in Figure 1 and Figure 2. The first step is the synthesis of NHC precursors (L1, L2, and L3): a solution of imidazole, 4,5-dichloroimidazole or benzimidazole was reacted with phenylethylene oxide in dry acetonitrile to obtain the corresponding N-alkylated products (P1, P2 and P3). The imidazolium salts were achieved by reaction of P1, P2, or P3 with sodium hydride in dry THF, followed by the addition of iodomethane. L1, L2, and L3 salts were isolated as white powders in good yields (42%, 59%, and 43%, respectively) and fully characterized by ^1^H-and ^13^C-NMR, and mass spectroscopy (see the Materials and Methods section). In the ^1^H-NMR, the signal of the NC*H*N was observed as a narrow singlet at 9.08 ppm, 9.49 ppm, and 9.72 ppm, respectively [30]. Further confirmation of the formation of imidazolium salts L1, L2, and L3 was given by ^13^C-NMR spectra that show the characteristic singlet of N*C*HN at 136.96 ppm, 136.72 ppm, and 137.34 ppm, respectively.

The reaction of imidazolium salts (L1, L2 or L3) with Ag_2_O, followed by trans-metalation with [Ru(*p*-cymene)Cl_2_]_2_ in dry dichloromethane, produced the expected ruthenium complexes (**RANHC-I, -V** and **-VI**) as orange or dark orange air-stable solids in good yields (60–90%).

The complexes **RANHC-I, -V,** and **-VI** were fully characterized by ^1^H- and ^13^C-NMR analysis, mass spectroscopy (see Appendix A), and elemental analyses. The ^1^H spectra confirm the successful complexes formation, because the signals of the starting imidazolium salt due to NC*H*N disappear, while in the ^13^C spectra, the signals attributable to carbene carbon (N*C*N) of **RANHC-I, -V,** and **-VI** at 174.24, 179.27 and 191.20 ppm, respectively, appear. The success of the synthesis of the complexes is confirmed by ESI-MS and elemental (C, H and N) analyses (see the Materials and Methods section). ESI-MS spectra show the peaks due to the [Ru(NHC)(*p*-cymene)Cl]^+^ species.

To produce complexes with the same ancillary ligands, but with different chemical-physical properties, we replaced the chloride ligands with different anions, i.e., (i) iodide, to have a more lipophilic complex [31,32]; (ii) PF_6_^−^, to have a more soluble complex in the physiological environment [33]; (iii) oxalate, a chelating agent, to have a thermodynamically more stable complex [24,34,35] (Figure 2).

**RANHC-II,-III,-IV** were obtained starting from **RANHC-I**. The iodide **RANHC-II** complex was generated in situ by adding NaI in dry THF for 12 h to **RANHC-I**, and complex **RANHC-III** was synthesized through the reaction between **RANHC-I** and AgPF_6_ for 4 h in dichloromethane dry. The complex **RANHC-IV** was obtained by reacting **RANCH-I** and Ag_2_C_2_O_4_ in dry THF for 12 h.

^1^H-NMR, ^13^C-NMR, mass
spectrometry, and elemental analyses, reported in the Appendix A,
confirm the synthesis of complexes **RANHC-II, -III** and **-IV**. The ^1^H-NMR
spectra show the peak relative to the CH of the *para*-cymene group at
3.16, 3.02, and 2.65 ppm, respectively. The ^13^C analysis shows the
signal relative to arbonic carbon at 171.01, 173.54, and 174.81, for **RANHC-II,
-III**, and **-IV**, respectively. The structure of complex **RANHC-IV**
was also confirmed by ^19^F-NMR and ^31^P-NMR analyses (see the
Materials and Methods section and Appendix A).

The proposed structures and hydrolytic stability were supported by conductivity measurements (solutions about 10^−3^ M); in fact, the conductance values for the Ru compounds I, IV, V, and VI determined in DMSO/H_2_O 90/10 are substantially independent of the concentration, which is close to zero. Instead, as expected, complex III showed a concentration-dependence conductivity in the range of 6.3–25.6 μS cm^−1^, confirming the electrolytic nature of the complex. Surprisingly, complex II also showed a concentration-dependent conductivity (in the range of 13.1–44.2 μS cm^−1^). This was attributed to the presence of a small amount of sodium iodide in the sample (about 10 mol%).

### 2.2. Anticancer Activity

The anticancer activity of all the synthesized Ru-NHC complexes (**RANHC I-VI**) was studied in two human breast cancer cell lines, namely the estrogen receptor positive (ER+) MCF-7 cells and the triple negative MDA-MB-231 cells (ER-, PR-, and HER-2/Neu not overexpressed), and neuroblastoma cells, i.e., SH-SY5Y. First, the cells were exposed for 72 h at different concentrations of the studied complexes (from 0.1 to 100 µM) and then their viability was checked by MTT assay.

The outcomes, summarized in Table 1 and Appendix A, evidenced that complexes **RANHC-V** and **VI** were the most active, being able to diminish the viability of all the cancer cells used in this assay. In particular, complexes **RANHC- V** and -**VI** resulted in more activity towards the aggressive and metastatic MDA-MB-231 cells, with IC_50_ values of 24.14 ± 0.7 and 40.57 ± 1.1 µM, respectively. Comparable activity was recorded as well against the ER(+) MCF-7 cells, with IC_50_ values of 26.05 ± 0.9 (**RANHC-V**) and 54.75 ± 1.1 (**RANHC-VI**) µM. Their anticancer activity against the breast cancer cells was comparable to that exerted by Cisplatin, used as the reference molecule (IC_50_ 32.15 ± 1.0 and 26.19 ± 1.1 µM against MDA-MB-231 and MCF-7, respectively). Instead, lower anticancer activity was recorded against the SH-SY5Y neuroblastoma cells, where complexes **RANHC-V** and **RANHC-VI** reported IC_50_ values of 48.43 ± 0.8 and 66.86 ± 0.8 µM, respectively.

The other Ru-NCH complexes showed IC_50_ values superior to 100 µM against all the cancer cells tested, with the exception of complexes **RANHC-I** and **-IV** that demonstrated a slight anticancer activity on neuroblastoma SH-SY5Y cells with IC_50_ values of 90.05 ± 1.2 and 88.89 ± 0.9 µM, respectively.

Additionally, we investigated the selectivity of all the synthesized Ru-NHC complexes on the cancer cells by studying their capability to interfere with the growth of two non-tumoral cells, namely the MCF-10A and BALB/3T3 cells. Among them, complexes **RANHC-V** and -**VI** exhibited a mild toxicity against the MCF-10A, with IC_50_ values of 79.47 ± 1.2 and 90.72 ± 1.2 µM, respectively; however, the complex **RANHC-V** was about 3.3-fold more active against the MDA-MB-231 than MCF-10A cells. Most importantly, complex **RANHC-V** was non-cytotoxic against the mouse embryonic fibroblasts BALB/3T3, at least until the concentration of 100 µM and under the adopted experimental conditions, contrarily to Cisplatin that exerted a high cytotoxicity (IC_50_ = 21.57 ± 1.2 µM). The complex **RANHC-VI**, instead, showed a higher cytotoxicity on both the non-tumoral cell lines used, with IC_50_ values of 39.09 ± 1.1 µM (BALB/3T3) and 90.72 ± 1.2 µM (MCF-10A). However, it was 2.2-fold more active on MDA-MB-231 cells than MCF-10A and less cytotoxic on BALB/3T3 than Cisplatin. The other Ru-NCH complexes showed IC_50_ values higher than 100 µM against both the adopted normal cells. Similar published complexes showed anticancer activity against different cancer cells, but with higher IC_50_ values or higher cytotoxic effects against the adopted normal cell lines [36,37].

Considering the obtained results, the presence of the two chlorine substituents on the imidazole ring at the positions 4 and 5, as for the complex **RANHC-V**, seemed to improve the anticancer activity, resulting in the most activity against the cancer cells used in our test. These outcomes are in agreement with our previous studies relative to the anticancer evaluation of some NHC metal complexes based on gold and silver atoms [17,18], in which the Au or Ag-NCH complexes with the chlorine substituents on the imidazole scaffold possessed better anticancer activity. Additionally, the fusion of a benzene ring on the imidazole group, as in complex **RANHC-VI**, increased the anticancer activity, as well, even if to a lesser extent. However, the benzimidazole scaffold seems to be also responsible for the mild cytotoxicity on the normal cells, notably on the mouse embryonic fibroblasts BALB/3T3.

Thus, we investigated the mechanism of action of the Ru-NHC complexes **RANHC-V** and **-VI**, introducing two chlorine substituents or a benzene ring, respectively, into the imidazole group, which exhibited the best anticancer activity.

It is well-established that NHCs were revealed as effective scaffolds able to interfere with several important biomolecules playing a major role in cancer onset and progression [38]. In our preceding studies conducted on gold and silver NHC complexes [17], we demonstrated that some of them resulted in good inhibitors of two enzymes, namely human topoisomerases I and II (hTopo I and II), which control and regulate the topological structure of DNA during essential cellular processes, including replication, transcription, duplication, etc. This classification depends on the different mechanism of action of these two enzymes. hTopo I is able to break one strand of the double helix, regulating the levels of DNA supercoiling, while hTopo II breaks both strands [39]. Topoisomerases have assumed primary importance in oncological research since tumor tissues present a higher concentration of these enzymes, thus finding new agents able to specifically target topoisomerases is an important goal that could be exploited for anticancer therapeutic purposes [40,41].

Based on these considerations, we studied the capability of the most active complexes **RANHC-V** and -**VI** in interfering with the hTopo I and II activity by the means of specific in vitro assays.

We performed the hTopo I relaxation assay using as a substrate the supercoiled plasmid pHOT1 and exposing the enzyme to our selected complexes or vehicle. The results, reported in Figure 3, clearly indicate that both complexes **RANHC-V** and -**VI** were able to totally inhibit the hTopo I activity at the concentration of 10 μM. Indeed, in lanes 4 and 5, corresponding to the enzymatic reactions in the presence of complexes **RANHC-V** or -**VI**, only the supercoiled DNA was present, noticeable as a single band in the lower portion of the agarose gel. Contrarily, in the CTRL (vehicle) reaction (Figure 3, lane 3), multiple bands related to the relaxed DNA products were detected.

Conversely, a different behavior was noticed in the hTopo II decatenation assay, which clearly demonstrated the absence of any inhibitory activity of the complexes **RANHC-V** and -**VI** used at two different concentrations, namely 10 and 50 μM. In Figure 4, only the results obtained at 50 μM concentration are reported (the same result was obtained at 10 μM) and as visible; under **RANHC-V** and -**VI** exposure, two bands related to the decatenation products at the bottom of the gel were detected (Figure 4, lanes 5 and 6), similarly to the CTRL reaction (Figure 4, lane 4). 

Thus, Ru-NHC complexes **RANHC-V** and -**VI** were proven to be selective inhibitors of the hTopo I activity at the concentration of 10 µM.

Topoisomerases are essential biological targets involved in several types of cancer progression, including breast cancer, and their inhibition induces DNA damage, triggering the programmed cell death [42].

Thus, using the TUNEL assay we determined whether the Ru-NHC complexes **RANHC-V** and -**VI** were able to trigger apoptosis in the MDA-MB-231 cells. The latter were first exposed to the Ru complexes, used at their IC_50_ values for 24 h, and then subjected to the rTdT enzyme, as indicated in the experimental section.

As follows from Figure 5, the breast cancer cells treated with complexes **RANHC-V** or -**VI** were TUNEL-positive; indeed, a green nuclear fluorescence associated with the DNA fragments formation was detected (Figure 5, panels B, **RANHC-V** and -**VI**). Conversely, in the vehicle (DMSO) treated cells, the lack of DNA damage was confirmed by the absence of fluorescence (Figure 5, panel B, CTRL).

### 2.3. Antibacterial Activity

The antibacterial activity of all the synthesized Ru-NHC complexes (**RANHC-I**-**VI**) was investigated against a Gram(-) bacterial strain, *Escherichia coli*, and two Gram(+) bacterial strains, *Enterococcus faecalis* and *S. aureus*. The obtained results for all the complexes in terms of minimum inhibitory concentration (MIC), expressed in µg/mL, are listed in Table 2 and Appendix A and, as it follows, all the tested complexes exhibited a higher antibacterial activity against the *S. aureus* Gram-positive strain, with a MIC of 25 µg/mL. The same result was obtained for complexes **RANHC-III** and **-VI** against the *E. coli* Gram(+) strain; indeed, they possessed MIC values of 25 µg/mL, as well. Instead, higher MIC values were recorded for the other complexes towards the same strain (MIC = 50 µg/mL for complexes **RANHC-I**, **-II**, **-IV,** and **-V**). Conversely, higher concentrations of the tested Ru complexes were necessary for inhibiting the *E. faecalis* strain growth. Specifically, complex **RANHC-V** inhibited *E. faecalis* at the concentration of 70 µg/mL, while all the other complexes exerted their inhibitory growth effects at the concentration of 50 µg/mL. DMSO, used as the vehicle, had no antimicrobial activity; in contrast, all the strains were ampicillin-sensible. We also determined the minimum bactericidal concentration (MBC), resulting in greater than 100 µg/mL for all the complexes. Comparable or lower MIC values against *S. aureus* were recently published for similar complexes, even though they were quite toxic against the HepG2 cells [37].

Overall, our results indicated the complexes **RANHC-III** and **-VI** were the best antibacterial agents against all the used strains, and that the *S. aureus* Gram-positive strain was the most sensitive to all the complexes.

### 2.4. Antioxidant Activity

The antioxidant effect of the synthesized Ru-NHC complexes was assessed by using the 2,2-diphenyl-1-picrylhydrazyl (DPPH) and the 2,20-azino-bis(3-ethylbenzo-thiazoline-6-sulphonic acid) (ABTS) radicals scavenging assays.

Generally, the DPPH radical is able to accept hydrogen radicals or an electron, turning into a stable molecule. The interaction with Ru-NHC complexes induces the reduction of DPPH radical intensity measured at 517 nm, due to the conversion of the purple DPPH radical into its corresponding yellow hydrazine form [43].

The obtained DPPH% scavenging activity for each Ru-NHC complex is shown in Figure 6, while the calculated IC_50_ values are listed in Table 3 and Appendix A.

The DPPH scavenging activity exerted by most of the Ru-NHC complexes was lower than that of Trolox, used as standard antioxidant in our assay, with the exception of complex **RANHC-III**, which exhibited a stronger electron-donating power than Trolox. In particular, complex **RANHC-III** and Trolox showed IC_50_ values of 44.19 ± 1.2 and 99.91 ± 0.9, respectively, whereas the other complexes produced IC_50_ values ranging from 161.6 ± 1.0 to 512.3 ± 0.8 µM (see Table 3). Thus, complex **RANHC-III** showed the highest DPPH scavenging activity among all the investigated complexes, whose scavenging ability against the DPPH radical follows the order: **RANHC-III** > **-VI** > **-II** > **-V** > **-I** > **-IV**.

To further analyze the Ru-NHC complexes antiradical potential, we examined their capability to interfere with the ABTS radical cation, a well-known protonated radical with a characteristic maximal absorbance at 730 nm, which diminishes with the scavenging of the radical proton. The assay measures radical scavenging by the electron donation [43]. The outcomes in terms of ABTS% scavenging activity are reported in Figure 7 and IC_50_ values of Ru-NHC complexes, together with the standard Trolox, on ABTS radical are summarized in Table 3 and Appendix A.

Surprisingly, all the Ru-NHC complexes exhibited higher ABTS scavenging activity than Trolox, which showed an IC_50_ value of 92.30 ± 0.9 µM. As for the DPPH assay, complex **RANHC-III** exhibited the highest ABTS scavenging activity, showing an IC_50_ value of 5.53 ± 1.1 µM.

Good antioxidant ability was also recorded under the complex **RANHC-IV** exposure, which showed an IC_50_ value of 8.57 ± 1.1 µM. The other complexes showed higher IC_50_ values, ranging from 11.36 ± 0.8 µM (for complex **RANHC-V**) to 17.21 ± 1.1 µM (for complex **RANHC-VI**) (see Table 3). Overall, the ABTS scavenging activity pattern of the Ru-NHC complexes can be graded in the following order: **RANHC-III** > **-IV** > **-V** > **-I** > **-II** > **-VI**. These activities are higher than those reported in studies on similar complexes [37].

The obtained results indicated that the antioxidant properties of all the Ru-NHC complexes and Trolox are concentration-dependent, as shown in Figure 4 and Figure 5. In particular, the tested complexes exhibited a higher ability in inhibiting the ABTS^•+^ than DPPH, showing better antioxidant activity against the latter radical cation than the well-known antioxidant Trolox.

## 3. Materials and Methods

### 3.1. Chemistry

All reactions were conducted under an N_2_ atmosphere using standard Schlenk and glovebox techniques. Reagents were obtained from Sigma Aldrich and TCI Chemicals and used without further purifications. Reactions involving silver-oxide were carried out in the dark. The solvents were dehydrated and deoxygenated under a nitrogen atmosphere, by heating under reflux over drying agents. NMR solvents (Euriso-Top products) were stored in the dark over molecular sieves.

The NMR spectra were recorded on a Bruker AM 250 spectrometer (250 MHz for ^1^H; 62.5 MHz for ^13^C), a Bruker AM 300 spectrometer (300 MHz for ^1^H; 75 MHz for ^13^C), and a Bruker AVANCE 400 spectrometer (400 MHz for ^1^H; 100 MHz for ^13^C; 161.97 MHz for ^31^P; 376 MHz for ^19^F). NMR samples were prepared by solubilizing about 15 mg of compound in 0.5 mL of deuterated solvent. The ^1^H and ^13^C NMR chemical shifts are referenced to SiMe_4_ (δ = 0 ppm) using the residual proton impurities of the deuterated solvents as internal standards. ^1^H NMR spectra are referenced using the residual solvent peak δ 2.50 for DMSO-d_6_ and δ 7.27 for CDCl_3_. ^13^C NMR spectra are referenced using the residual solvent peak at δ 39.51 for DMSO-d_6_ and δ 77.23 for CDCl_3_. Multiplicities are abbreviated as follows: singlet (s), doublet (d), triplet (t), multiplet (m), doublet of doublets (dd), broad (br), and overlapped (o). Elemental analysis was conducted using a PERKIN-Elmer 240-C analyzer. ESI-MS measurements were performed on a Waters Quattro Micro triple quadrupole mass spectrometer, equipped with an electrospray ion source. MALDI-MS mass spectra were achieved by a Bruker SolariX XR Fourier transform ion cyclotron resonance mass spectrometer (Bruker Daltonik GmbH, Bremen, Germany) with a 7 T refrigerated actively shielded superconducting magnet (Bruker Biospin, Wissembourg, France). MALDI ion source (Bruker Daltonik GmbH, Bremen, Germany) was used in the samples in positive ion mode. The mass range was set to *m*/*z* 200–3000. The laser power was 28% and 22 laser shots were utilized for each scan. The mass spectra were calibrated externally using a mix of peptide clusters in MALDI ionization positive ion mode. A linear calibration was applied. The molar conductance of the ruthenium complexes was measured at room temperature by a Metrohm 712 Conductometer using freshly prepared 10^−3^ M solutions in DMSO/H_2_O 90/10 solvent.

#### 3.1.1. General Procedure for Synthesis of *N*-Heterocyclic Carbene Proligands (**L1**, **L2** and **L3**)

Imidazolium salts **L1**, **L2** and **L3** were synthesized using the synthetic procedure reported in related studies [14,18,30,44].

First step: in a round bottom flask equipped with a magnetic stirrer and condenser, a solution of imidazole, 4,5-dichloroimidazole or benzimidazole (1.0 eq) was reacted with phenylethylene oxide (1.2 eq) in dry acetonitrile for 12 h at the refluxing temperature to obtain the corresponding *N*-monoalkylated products (**P1, P2,** and **P3**) as white powders.

Second step: sodium hydride (1.7 eq) was introduced in the solutions of **P1**, **P2,** or **P3** (1.0 eq) in dry THF and the mixtures were stirred for 2 h at 4 °C. Then, iodomethane was added to the solutions, which were stirred for a further 12 h at room temperature.

The imidazolium salts **L1**, **L2,** and **L3** were recovered as white powders removing the solvent and washing with hexane (3 × 20 mL) and diethyl ether (3 × 20 mL). Yields: **L1** 42%, **L2** 59%, and **L3** 43%.

##### Characterization of *N*-Methyl, *N′*-(2-Methoxy-2-phenyl)ethyl Imidazolium Iodide (**L1**)

^1^H-NMR (ppm, DMSO-d_6_, 250 MHz): δ 9.08 (s, NC***H***N, 1H), 7.70–7.38 (m, aromatic hydrogens and NC***H***C***H***N, 7H), 4.66 (m, C***H***OCH_3_, 1H), 4.41 (m, NC***H*_2_**CHOCH_3_, 2H), 3.88 (s, NC***H*_3_**, 3H), 3.13 (s, OC***H*_3_**, 3H) (Appendix A).

^13^C-NMR (ppm, DMSO-d_6_, 62.5 MHz): δ 137.23 (*ipso* carbon of aromatic ring), 136.96 (N***C***HN), 128.67, 128.55 and 126.71 (aromatic carbons), 123.18, 122.97 (N***C***H***C***HN), 80.44 (***C***HOCH_3_), 56.36 (N***C***H_2_CHOCH_3_), 53.75 (O***C***H_3_), 35.80 (N***C***H_3_) (Appendix A).

Elemental analysis: calcd. for C_13_H_17_IN_2_O: C, 42.36; H, 4.98; N, 8.14. Found: C, 42.30; H, 4.92; N, 8.10.

Maldi-MS (CH_3_CN) calcd/found (*m*/*z*): [C_13_H_17_N_2_O]^+^ 217.1335/217.1333 (Appendix A).

##### Characterization of *N*-Methyl, *N′*-(2-Methoxy-2-phenyl)ethyl-4,5-Dichloro Imidazolium Iodide (**L2**)

^1^H-NMR (ppm, DMSO-d_6_, 400 MHz): δ 9.49 (s, NC***H***N, 1H), 7.45–7.37 (m, aromatic hydrogens, 5H), 4.64 (t, C***H***OCH_3_, 1H), 4.46 (m, NC***H*_2_**CHOCH_3_, 2H), 3.89 (s, OC***H*_3_**, 3H), 3.15 (s, NC***H*_3_**, 3H) (Appendix A).

^13^C-NMR (ppm, DMSO-d_6_, 100 MHz): δ 136.72 (*ipso* carbon of aromatic ring), 136.72 (N***C***HN), 128.90, 126.79, 126.00 (aromatic carbons), 119.09, 118.85 (N***C***H***C***HN), 79.57 (***C***HOCH_3_), 56.50 (N***C***H_2_CHOCH_3_), 52.91 (O***C***H_3_), 35.20 (N***C***H_3_) (Appendix A).

Elemental analysis: calcd. for C_13_H_15_Cl_2_IN_2_O: C, 37.80; H, 3.66; N, 6.78. Found: C, 37.75; H, 3.61; N, 6.73.

Maldi-MS (CH_3_CN) calcd/found (*m*/*z*): [C_13_H_16_Cl_2_N_2_O]^+^ 285.0556/285.0544 (Appendix A).

##### Characterization of *N*-Methyl, *N′*-(2-Methoxy-2-phenyl)ethyl-Benzoimidazolium Iodide (**L3**)

^1^H-NMR (ppm, DMSO-d_6_, 250 MHz): δ 9.72 (s, NC***H***N, 1H), 8.02–7.42 (m, aromatic hydrogens, 9H), 4.76 (o, C***H***OCH_3_ and NC***H*_2_**CHOCH_3_, 3H), 4.13 (s, OC***H*_3_**, 3H), 3.09 (s, NC***H*_3_**, 3H) (Appendix A).

^13^C-NMR (ppm, DMSO-d_6_, 75 MHz): δ 143.21 (*ipso* carbon of aromatic ring), 137.34 (N***C***HN), 131.41 128.72, 127.00, 126.46, 126.38, 113.98, 113.50 (aromatic carbons), 80.22 (***C***HOCH_3_), 56.39 (N***C***H_2_CHOCH_3_), 51.55 (O***C***H_3_), 33.39 (N***C***H_3_) (Appendix A).

Elemental analysis: calcd. for C_17_H_19_IN_2_O: C, 51.79; H, 4.86; N, 7.11. Found: C, 51.75; H, 4.82; N, 7.07.

Maldi-MS (CH_3_CN) calcd/found (*m*/*z*): [C_17_H_19_N_2_O]^+^ 267.1492/267.1478 (Appendix A).

#### 3.1.2. General Procedure of the Synthesis of Ruthenium Complexes (**RANHC-I, V,** and **VI**)

Ruthenium complexes **RANHC-I, -V,** and **-VI** were synthesized using a procedure reported in a previous study, modified for our purposes [27].

The imidazolium salts (**L1** or **L2** or **L3-**2.00 eq) were dissolved in dry dichloromethane in a round bottom flask. Ag_2_O (1.00 eq) was introduced to obtain the silver complexes and the mixtures were refluxed in the dark for 12 h. Then, [Ru(*p*-cymene)Cl_2_]_2_ (1.00 eq) was added to the mixtures, and the resulting suspensions were left to reflux for another 12 h. Afterwards, they were filtered and concentrated, and the complexes were recovered as brownish-orange powder by precipitation with *n*-hexane. Yields: **RANHC-I** 60%, **RANHC-V** 63%, and **RANCH-VI** 90%.

##### Characterization of **RANHC-I**

^1^H-NMR (ppm, CDCl_3_, 300 MHz): δ 7.60–6.96 (m, hydrogens on aromatic ring and NC***H***C***H***N, 7H), 5.38–5.30 (dd, aromatic hydrogens of *p*-cymene, 2H), 5.03 (m, aromatic hydrogens of *p*-cymene, 2H), 4.77 (m, NC***H*_2_**CHOCH_3_, 2H), 4.02 (o, OC***H_3_
***and NCH_2_C***H***OCH_3_, 4H), 3.17 (s, NC***H_3_***_,_ 3H ), 2.86 (m, C***H***(CH_3_)_2_
*p*-cymene, 1H), 1.97 (s, C***H_3_
****p*-cymene, 3H), 1.22 (m, CH(C***H_3_***)_2_ *p*-cymene, 6H) (Appendix A).

^13^C-NMR (ppm, CDCl_3_, 62.5 MHz): δ 174.24 (N***C***N), 138.99 (*ipso* carbon of aromatic ring), 128.45, 127.96, 127.24, 123.11 (aromatic carbons), 108.67, 99.05 (N***C***H***C***HN), 85.31, 85.16, 83.35, 82.56, 82.32 (aromatic carbons *p*-cymene, NCH_2_***C***H), 56.92 (N***C***H_2_CH), 56.41 (O***C***H_3_), 39.58 (***C***H(CH_3_)_2_ *p*-cymene), 30.64 (N***C***H_3_), 22.52, 22.30 (CH(***C***H_3_)_2_ *p*-cymene), 18.44 (***C***H_3_ *p*-cymene) (Appendix A).

Elemental analysis: calcd. for C_23_H_30_Cl_2_N_2_ORu: C, 52.87; H, 5.79; N, 5.36. Found: C, 52.80; H, 5.71; N, 5.29. ESI-MS (CH_2_Cl_2_) calcd/found (*m*/*z*): [C_23_H_30_N_2_OClRu]^+^ 487.1088/487.1085 (Appendix A).

##### Characterization of **RANHC-V**

^1^H-NMR (ppm, CDCl_3_, 300 MHz): δ 7.41–7.34 (m, hydrogens on aromatic ring, 5H), 5.84–4.85 (o, aromatic hydrogens *p*-cymene-NCH_2_C***H***-NC***H_2_***CH, 7H), 4.03 (s, OC***H_3_***, 3H), 3.24 (s, NC***H_3_***, 3H), 2.99 (m, C***H***(CH_3_)_2_
*p*-cymene, 1H), 2.16 (s, C***H_3_*** *p*-cymene, 3H), 1.29 (m, CH(C***H_3_***)_2_, 6H) (Appendix A).

^13^C-NMR (ppm, CDCl_3_, 75 MHz): δ 179.27 (N***C***N), 137.39 (*ipso* carbon of aromatic ring), 128.66, 128.35, 125.94 (aromatic carbons), 118.54, 116.29 (N***C***H***C***HN), 108.26, 99.95, 86.74 (aromatic carbons *p*-cymene), 85.07 (NCH_2_***C***H), 83.31, 80.50 (aromatic carbons *p*-cymene), 56.67 (N***C***H_2_CH), 56.48 (O***C***H_3_), 38.46 (N***C***H_3_), 30.36 (***C***H(CH_3_)_2_ *p*-cymene), 23.91, 20.57 (CH(***C***H_3_)_2_ *p*-cymene), 18.38 (***C***H_3_ *p*-cymene) (Appendix A).

Elemental analysis: calcd. for C_23_H_28_Cl_4_N_2_ORu: C, 46.71; H, 4.77; N, 4.74. Found: C, 46.65; H, 4.71; N, 4.68. ESI-MS (CH_2_Cl_2_) calcd/found (*m*/*z*): [C_23_H_28_RuN_2_OCl_3_]^+^ 557.0292/557.0334 (Appendix A).

##### Characterization of **RANHC-VI**

^1^H-NMR (ppm, CDCl_3_, 400 MHz): δ 7.67–7.32 (m, hydrogens of aromatic rings, 9H), 5.77, 5.56 (br, hydrogens *p*-cymene, 2H), 5.48 (br, NCH_2_C***H***, 1H), 5.17 (m, NC***H_2_***CH, 2H), 5.05, 4.50 (d, hydrogens *p*-cymene, 2H), 4.25 (s, OC***H_3_***, 3H), 3.06 (o, NC***H_3_
***and C***H***(CH_3_)_2_
*p*-cymene, 4H), 2.09 (s, C***H_3_*** *p*-cymene, 3H), 1.30 (m, CH(C***H_3_***)_2_, 6H) (Appendix A).

^13^C-NMR (ppm, CDCl_3_, 75 MHz): δ 191.20 (N***C***N), 138.41 (*ipso* carbon of aromatic ring), 135.71, 134.65, 128.50, 128.09, 126.44, 122.49, 122.30, 111.38, 109.52 (aromatic carbons), 109.19, 99.84, 87.09, 85.72 (aromatic carbons of *p*-cymene), 83.34 (NCH_2_***C***H), 81.24, 81.13 (aromatic carbons of *p*-cymene), 56.20 (O***C***H_3_), 55.49 (N***C***H_2_CH), 36.52 (***C***H(CH_3_)_2_ *p*-cymene), 30.36 (N***C***H_3_), 23.39, 21.00 (CH(***C***H_3_)_2_ *p*-cymene), 18.26 (***C***H_3_, *p*-cymene) (Appendix A).

Elemental analysis: calcd. for C_27_H_32_Cl_2_N_2_ORu: C, 56.64; H, 5.63; N, 4.89. Found: C, 56.60; H, 5.61; N, 4.80. ESI-MS (CH_2_Cl_2_) calcd/found (*m*/*z*): [C_27_H_32_ClN_2_ORu]^+^ 537.1244/537.1272 (Appendix A).

#### 3.1.3. Synthesis of **RANHC-II**

In a round bottom flask, complex **RANHC-I** (100.0 mg, 0.19 mmol, 1.00 eq) and NaI (142.4 mg, 0.95 mmol, 5.00 eq) were mixed in dry THF (10 mL). The mixture was stirred for 8 h at room temperature. The resulting suspension was filtered through a plug of celite and the dark orange solution was concentrated under vacuum to yield the complex **RANHC-II** as dark orange powder (43% yield) [31,32].

^1^H-NMR (ppm, CDCl_3_, 400 MHz): δ 7.67–7.03 (m, hydrogen of aromatic ring and NC***H***C***H***N, 7H), 5.68–5.42 (br, aromatic hydrogens of *p*-cymene, 3H), 5.17 (o, aromatic hydrogens of *p*-cymene and NCH_2_C***H***OCH_3_, 2H), 4.81 (m, NC***H*_2_**CHOCH_3_, 2H), 4.13 (s, OC***H_3_***, 3H), 3.16 (o, NC***H_3_
***and C***H***(CH_3_)_2_
*p*-cymene, 4H ), 1.88 (s, C***H_3_
****p*-cymene, 3H), 1.22 (m, CH(C***H_3_***)_2_ *p*-cymene, 6H) (Appendix A).

^13^C-NMR (ppm, CDCl_3_, 100 MHz): δ 171.01 (N***C***N), 138.00 (*ipso* carbon of aromatic ring), 128.15, 127.04 (aromatic carbons), 123.49,123.09 (N***C***H***C***HN), 103.81, 97.09 (aromatic carbons *p*-cymene), 86.03 (NCH_2_***C***H), 82.35, 82.07, 81.58, 81.48 (aromatic carbons *p*-cymene), 59.20 (N***C***H_2_CH), 56.01 (O***C***H_3_), 44.74 (***C***H(CH_3_)_2_ *p*-cymene), 31.06 (N***C***H_3_), 23.59, 21.33 (CH(***C***H_3_)_2_ *p*-cymene), 18.32 (***C***H_3_ *p*-cymene) (Appendix A).

Elemental analysis: calcd. for C_23_H_30_I_2_ON_2_Ru: C, 39.16; H, 4.29; N, 3.97. Found: C, 39.13; H, 4.26; N, 3.94. ESI-MS (CH_2_Cl_2_) calcd/found (*m*/*z*): [C_23_H_30_ION_2_Ru]^+^ 579.0447/579.0450 (Appendix A).

#### 3.1.4. Synthesis of **RANHC-III**

AgPF_6_ (48.0 mg, 0.19 mmol, 1.00 eq), **RANHC-I** (100.0 mg, 0.19 mmol, 1.00 eq) and dry CH_2_Cl_2_ (7 mL) were introduced in a round bottom flask and stirred for 20h at room temperature. The resulting suspension was filtered through celite and the orange solution was concentrated *in vacuo* to yield complex **RANHC-III** as orange powder (50% yield) [33].

^1^H-NMR (ppm, CDCl_3_, 300 MHz): δ 7.43–6.53 (m, hydrogen of aromatic ring and NC***H***C***H***N, 7H), 5.78–4.08 (o, aromatic hydrogens of *p*-cymene-NC***H*_2_**CHOCH_3_-NCH_2_C***H***OCH_3_, 7H), 3.87 (s, OC***H_3_***, 3H), 3.80 (s, C***H_3_
****p*-cymene, 3H), 3.02 (m_,_ C***H***(CH_3_)_2_
*p*-cymene, 1H ), 2.27 (s, NC***H_3_***, 3H), 1.28 (m, CH(C***H_3_***)_2_ *p*-cymene, 6H) (Appendix A).

^13^C-NMR (ppm, CDCl_3_, 62.5 MHz): δ 173.54 (N***C***N), 133.21 (*ipso* carbon of aromatic ring), 128.29, 126.20 (aromatic carbons), 123.92, 123.00 (N***C***HN***C***HN), 113.64, 100.64 (aromatic carbons *p*-cymene), 81.95 (aromatic carbon *p*-cymene), 81.79 (NCH_2_***C***H), 81.10, 78.38, 78.16 (aromatic carbons *p*-cymene), 66.83 (O***C***H_3_), 55.97 (N***C***H_2_CH), 37.60 (***C***H(CH_3_)_2_ *p*-cymene), 31.16 (***C***H_3_ *p*-cymene), 30.88 (N***C***H_3_), 23.32, 20.00 (CH(***C***H_3_)_2_ *p*-cymene) (Appendix A).

^31^P-NMR (ppm, CDCl_3_, 161.97 MHz): δ -143.97 (m) (Appendix A).

^19^F-NMR (ppm, CDCl_3_, 376 MHz): δ -71.64, -73.54 (Appendix A).

Elemental analysis: calcd. for C_23_H_30_ClF_6_N_2_OPRu: C, 43.71; H, 4.78; N, 4.43. Found: C, 43.68; H, 4.23; N, 4.40. ESI-MS (CH_2_Cl_2_) calcd/found (*m*/*z*): [C_23_H_30_ClN_2_ORu]^+^ 487.1088/487.1118 (Appendix A).

#### 3.1.5. Synthesis of **RANHC-IV**

Into a bottom flask equipped with a magnetic stirrer containing 27 mL of dry THF, **RANHC-I** (100.0 mg, 0.19 mmol, 1.00 eq) and 85.0 mg of Ag_2_C_2_O_4_ (0.28 mmol, 1.50 eq) were introduced. The reaction mixture was stirred at room temperature for 12h. The resulting suspension was filtered to remove the precipitate. The THF-soluble product was dried by evaporation of the solvent and **RANHC-IV** was yielded as a dark green solid (78% yield) [45].

^1^H-NMR (ppm, CDCl_3_, 400 MHz): δ 7.68–6.93 (m, aromatic hydrogens and NC***H***C***H***N, 7H), 5.46–5.34 (dd, aromatic hydrogens of *p*-cymene, 2H), 5.16–5.07 (dd, aromatic hydrogens of *p*-cymene, 2H), 4.47–4.27 (dd, NC***H_2_***CH, 2H), 3.98 (t, NCH_2_C***H***, 1H), 3.76 (s, OC***H_3_***, 3H), 3.15 (s, NC***H_3_***, 3H), 2.65 (m, C***H***(CH_3_)_2_ *p*-cymene, 1H), 1.98 (s, C***H_3_*** *p*-cymene, 3H), 1.18 (m, CH(C***H_3_***)_2_ *p*-cymene, 6H) (Appendix A).

^13^C-NMR (ppm, CDCl_3_, 62.5 MHz): δ 174.81 (N***C***N), 165.51, 165.41 (***C***=O, oxalyl group), 138.68 (*ipso* carbon of aromatic ring), 128.68, 128.13,127.10 (aromatic carbons), 123.41,123.04 (N***C***H***C***HN), 107.30, 97.18 (aromatic carbons *p*-cymene), 84.50 (NCH_2_***C***H), 83.64, 82.74, 82.13, 81.72 (aromatic carbons *p*-cymene), 56.67 (O***C***H_3_), 56.42 (N***C***H_2_CH), 37.41 (***C***H(CH_3_)_2_ *p*-cymene), 31.36 (N***C***H_3_), 22.65, 22.34 (CH(***C***H_3_)_2_ *p*-cymene), 18.36 (***C***H_3_ *p*-cymene) (Appendix A).

Elemental analysis: calcd. for C_25_H_30_N_2_O_5_Ru: C, 55.65; H, 5.60; N, 5.19. Found: C, 55.60; H, 5.55; N, 5.14. ESI-MS (CH_2_Cl_2_) calcd/found (*m*/*z*): [C_25_H_30_N_2_O_5_Ru]^+^ 541.1278/541.1248 (Appendix A).

### 3.2. Biology

#### 3.2.1. Cell Cultures

The adopted cells (breast cancer MCF-7 and MDA-MB-231 and human mammary epithelial MCF-10A cells) were purchased from American Type Culture Collection (ATCC, Manassas, VA, USA) and cultured as indicated [18]. The mouse embryonic fibroblast BALB/3T3 and neuroblastoma SH-SY5Y were also obtained from American Type Culture Collection (ATCC, Manassas, VA, USA) and cultured in DMEM high glucose supplemented with 100 U mL^−1^ penicillin/streptomycin and 10% bovine calf serum (BCS) or 10% fetal bovine serum (FBS), respectively.

#### 3.2.2. MTT Assay

MTT assay (Sigma Aldrich (St. Louis, MO, USA)) was used for the in vitro anticancer activity evaluation, as previously reported [46]. The Ru-NHC complexes were employed at different concentrations (0.1–1–10–25–50–100 μM) for 72 h. The IC_50_ values were calculated from the percent (%) of control using GraphPad Prism 9 (GraphPad Software, La Jolla, CA, USA).

#### 3.2.3. hTopo I Relaxation Assay and hTopo II Decatenation Assay

hTopoI relaxation assays were done using the supercoiled pHOT1 as substrate, with the recombinant hTopoI (TopoGEN, Port Orange, FL, USA) and compounds **RANHC**-**V** and -**VI** or DMSO (CTRL) following the manufacturer’s protocol (TopoGEN, Port Orange, FL, USA), with some changes [47].

Similarly, hTopoII decatenation assays were carried out incubating the kinetoplast DNA (kDNA) substrate with the hTopoII (TopoGEN, Port Orange, FL) and compounds **RANHC**-**V** and -**VI** or DMSO (CTRL) following the manufacturer’s procedures (TopoGEN, Port Orange, FL, USA), with some changes [47].

#### 3.2.4. TUNEL Assay

A TUNEL assay was used for cells apoptosis detection, following the manufacturer’s protocols (CF™488A TUNEL Assay Apoptosis Detection Kit, Biotium, Hayward, CA, USA), with some changes. Briefly, cells were seeded and then processed, as described [47]. DAPI (0.2 μg/mL, Sigma Aldrich, Milan, Italy) staining was adopted for nuclei. A fluorescence microscope (Leica DM 6000) was used for fluorescence detection (20x magnification). LAS-X software was used to acquire and process all the images that are representative of three different experiments.

#### 3.2.5. Minimum Inhibitory Concentration (MIC) and Minimum Bactericidal Concentration (MBC) Determination

One Gram-negative (*Escherichia coli* (ATCC^®^ 25922TM)) and two Gram-positive bacterial strains (*Enterococcus faecalis* (ATCC^®^ 19433TM) and *Staphylococcus aureus* (ATCC^®^ 23235TM)] were used for MIC and MBC values determination, according to CLSI guidelines [48].

MIC represents the lowest concentration of compound inhibiting the visible microbial growth and expressed as μg/mL; MBC is the lowest concentration able to kill the bacteria. Both determinations were performed by the broth dilution method.

Bacteria were grown overnight in LB medium (2%), then diluted to a density of 4000 colony forming units (CFUs) per mL, plated in sterile 96-well microplates, obtaining about 105 cells/ well. Increasing concentrations of the Ru-NHC complexes (1, 10, 25, 50, 70, 100 µg/mL) were used. After incubation at 37 °C for 18 h (overnight), bacterial growth was checked at a wavelength of 600 nm using a Multiskan spectrophotometer (model Multiskan Ex Microplate; Thermo Scientific, Nyon, Switzerland) and MIC or MBC values were determined, comparing cell density with a positive control (bacterial cells grown in LB medium were added with only the vehicle, DMSO). Each experiment was carried out five times, in triplicate. Ampicillin (Sigma Aldrich A9393) was used as the control for strain sensitivity.

#### 3.2.6. Antioxidant Activity

##### 2,2-Diphenyl-1-Picrylhydrazyl (DPPH) Assay

The radical scavenging properties of Ru-NHC complexes on the 1,1-diphenyl-2-picryl-hydrazil (DPPH) radical were evaluated as described by Fazio et al. [49], with minor corrections. Then, 20 μL of each Ru complex, properly dissolved in DMSO, were mixed with methanol DPPH 0.1 mM (180 μL) in a 48-well plate obtaining seven different concentrations (10, 50, 100, 200 and 300 µg/mL). Next, 20 μL of DMSO in 180 μL of DPPH methanol solution was diluted as the control. The mixture was vigorously shaken and incubated (room temperature, 30 min, in the dark). The scavenging activity was measured at 517 nm, using a microplate reader. DPPH radical scavenging was expressed as inhibition percentages (%I_DPPH_) for the Ru complexes compared to the initial concentration of DPPH (control) according to the formula:$$\%I_{\mathrm{DPPH}}=\left[\frac{A_{0}-A}{A_{0}}\right]\times 100$$
where *A*_0_ is the absorbance at 517 nm of the control reaction and *A* is the absorbance at 517 nm in the presence of samples.

The obtained I_DPPH_ percentages allowed the IC_50_ values determination, using GraphPad Prism 9 software (GraphPad Inc., San Diego, CA, USA). Trolox was employed as a positive control.

##### 2,2′-Azinobis(3-Ethylbenzothiazoline-6-Sulfonic Acid (ABTS) Assay

The radical scavenging properties of the Ru-NHC complexes on a 2,2′-azino-bis(3-ethylbenzothiazoline-6-sulfonate) radical cation (ABTS^•+^) were calculated as described by Fazio et al. [49], with few adjustments. Briefly, 2 mM ABTS and 70 mM potassium persulfate water solutions were mixed and incubated to obtain ABTS^•+^ radical stock solution (16 h, room temperature, in the dark). ABTS solution was diluted in ethanol to obtain an absorbance of 0.70 ± 0.02 at 730 nm, prior to use. Then, 2 μL of each Ru-NHC complex at different concentrations, properly dissolved in DMSO, were mixed with 198 μL of the ABTS^•+^ solution in a 48-well plate obtaining the following concentrations: 0.1, 1, 10, 50, and 100 µg/mL. The solutions were vigorously shaken and incubated (5 min, room temperature, in the dark). The ABTS scavenging activity was determined at 730 nm, using a microplate reader. The control was prepared mixing 2 μL of DMSO with 198 μL of the ABTS^•+^ solution. The ABTS radical scavenging was expressed as inhibition percentages (%I_ABTS_) of each complex compared to the control, according to the formula:$$\%I_{\mathrm{ABTS}}=\left[\frac{A_{0}-A}{A_{0}}\right]\times 100$$
where *A*_0_ is the absorbance at 730 nm of the control reaction and *A* is the absorbance at 730 nm in the presence of samples.

IC_50_ values for the Ru complexes were calculated from the %I_ABTS_ using GraphPad Prism 9 software (GraphPad Inc., San Diego, CA, USA). Trolox was employed as the positive control.

## 4. Conclusions

In recent years, the search for new anticancer compounds endowed with better selectivity and, consequently, fewer side effects embraced non-platinum compounds as, for instance, the Ru-based complexes Herein, we designed and synthesized a series of Ru-NHC complexes, stabilized by η^6^-arene coordinated ligand, namely RANHC complexes, and evaluated their anticancer, antimicrobial, and antioxidant properties. Our outcomes demonstrated that these new and interesting complexes possess good anticancer activity, mostly against MDA-MB-231 cells, and the ability to block hTopoI activity, triggering the cancer cells’ death by apoptosis. Moreover, we determined the MIC values on *E. coli*, *E. faecalis,* and *S. aureus*, which indicated antibacterial activity, particularly against *S. aureus*. Finally, the antioxidant properties were found to be better than those of Trolox, used as a reference, in scavenging the ABTS radical. Summing up, the new Ru complexes exhibit interesting properties, which deserve to be further deepened and developed because of their potential synergistic effect in the management of multifactorial diseases, as cancer, or in antimicrobial-resistant infections, exerting, at the same time, protection against the oxidative stress characterizing these pathological conditions.

## Data Availability

Not applicable.

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
