# Peer review of "Synthesis of Novel N-Heterocyclic Carbene-Ruthenium (II) Complexes, “Precious” Tools with Antibacterial, Anticancer and Antioxidant Properties"

_antibiotics, 2023, doi:10.3390/antibiotics12040693_

Round 1

Reviewer 1 Report

The paper entitled “Synthesis of novel NHC-Ruthenium (II) complexes, “precious” tools with antibacterial, anticancer and antioxidant properties” is an interesting piece of work that describes the synthesis and characterization of new R-NHC complexes with interesting cytotoxic activity as well as antibacterial activity that deserves publication.

Despite that there are some points to be improved before publication.

- All biological studies were performed in aqueous media (with complexes dissolved previously in DMSO). Therefore, the stability of the compounds in DMSO and DMSO/water mixtures need to be explored. At least it would be necessary to have an insight into it by registering the conductivity. If another methodology can be added it would be welcomed. It can also be at least discussed from bibliography.

-The results must be put into context, please compare the IC50 and MIC values with others of related complexes (those of the same metal, of other metals containing the same ligand). In order to easy comparisons, it would be advisable to express values both in micromolar and mg/mL.

Other minor points are included int the word file attached.

Author Response

Reviewer 1

-                                                                                                                                                                   with

 an interesting piece of work that describes the synthesis and characterization of new R-NHC complexes with interesting cytotoxic activity as well as antibacterial activity that deserves publication.

Despite that there are some points to be improved before publication.

- All biological studies were performed in aqueous media (with complexes dissolved previously in DMSO). Therefore, the stability of the compounds in DMSO and DMSO/water mixtures need to be explored. At least it would be necessary to have an insight into it by registering the conductivity. If another methodology can be added it would be welcomed. It can also be at least discussed from bibliography.

Authors reply: We thank the reviewer for the suggestion, this data can certainly improve the accuracy of the work.

Thus, we have performed measurements of molar conductance of the ruthenium complexes and added in the manuscript:

In the Experimental part:

The molar conductance of the ruthenium complexes was measured at room temperature by a Metrohm 712 Conductometer using freshly prepared 10-3 M solutions in DMSO/H2O 90/10  solvent.

In the Results:

The proposed structures and hydrolytic stability were supported by conductivity measurements (solutions about 10-3 M), in fact the conductance values for the Ru compounds I, IV, V and VI determined in DMSO/ H2O 90/10 are substantially independent of the concentration, it is close to zero. Instead, as expected, complexes III showed a concentration-dependence conductivity in the range of 6.3–25.6 μS cm-1, confirming the electrolytic nature of the complex. Surprisingly, complex II also showed a concentration-dependent conductivity (in the range of 13.1–44.2 μS cm-1), this was attributed to the presence of a small amount of sodium iodide in the sample (about 10 mol%).

-The results must be put into context, please compare the IC50 and MIC values with others of related complexes (those of the same metal, of other metals containing the same ligand). In order to easy comparisons, it would be advisable to express values both in micromolar and mg/mL.

Authors reply: As requested, the IC50 and MIC values of the reported complexes have been reported either as µM and µg/mL (see Table S1, S2 and S3 in the Supplementary materials). However, a general comparison with the huge number of published Ru-Complexes is not easy to do, mainly for the following reasons: ligands are not the same, cell/bacterial models are different, and even if the same cell line has been adopted, other variables strictly influence the IC50 and MIC values (growth condition, passage number, time of exposure, etc). Finally, many published papers lack the safety studies on normal cells. Thus, we added some of the above considerations in the discussion, regarding the most similar published complexes and conditions.

Other minor points: please revise the following expressions:

N-

Authors reply: Done, thanks.

Introduction:

Ru is an awesome catalyzer

Conclusions: Finally, the antioxidant properties were investigated, resulting better than the well-known Trolox in scavenging the ABTS radical

Authors reply: we edited the sentences.

Conclusions

The following belongs to the introduction not to the conclusions

anticancer compounds endowed with a better selectivity and, consequently, less side 658 effects embraced non-platinum compounds. Amongst different metals, the Ru-based an- 659 ticancer complexes showed a great potential as alternative to platinum-based compounds 660 and were demonstrated to be considerably promising as well as antibacterial, antioxidant 661 and anti-

Authors reply: The paragraph has been changed.

Reviewer 2 Report

The authors reported on the synthesis, characterization, and biological activity of new Ru complexes containing N-heterocyclic carbene ligands. Certainly, this manuscript should be of interest to a wide range of bio/organic chemists, but the work can be published only after some issues have been resolved.

1) As mentioned in Materials and Methods part, mass spectra were achieved by SolariX.  Ultrahigh-resolution mass spectra should be reported to 4 decimal, calculated value and the error places in the main body of the article. A comparison of the experimental spectrum with the calculated one and calculation of the error should be added to the SI as Figure for all spectra. 
2) It would be more convenient for the reader to place the yields of the complexes directly on Scheme 1. 

Author Response

Reviewer 2

The authors reported on the synthesis, characterization, and biological activity of new Ru complexes containing N-heterocyclic carbene ligands. Certainly, this manuscript should be of interest to a wide range of bio/organic chemists, but the work can be published only after some issues have been resolved.

  • As mentioned in Materials and Methods part, mass spectra were achieved by SolariX.  Ultrahigh-resolution mass spectra should be reported to 4 decimal, calculated value and the error places in the main body of the article. A comparison of the experimental spectrum with the calculated one and calculation of the error should be added to the SI as Figure for all spectra. 

Authors reply: The authors express their gratitude to the reviewer for his/her positive comments.

The revised version of the manuscript has been implemented reporting the mass spectra to 4 decimals and the calculated value. The Figures of experimental and calculated spectra have been added in the SI.

2) It would be more convenient for the reader to place the yields of the complexes directly on Scheme 1.

Authors reply: Done, thanks. We thank the reviewer for his/her suggestion, actually reporting the yields of the complexes directly on Scheme 1 makes it easier to read.

Round 2

Reviewer 1 Report

The paper has improved and is adequate for publication.

There are some minor typos to correct in the document attached.
